# Live well, die well – an international cohort study on experiences, concerns and preferences of patients in the last phase of life: the research protocol of the iLIVE study

Berivan Yildiz ![ORCID],[1] Simon Allan,[2] Misa Bakan,[3] Pilar Barnestein-Fonseca,[4,5] Michael Berger,[6] Mark Boughey,[7] Andri Christen,[8] Gustavo G De Simone,[9] Martina Egloff,[8] John Ellershaw,[10] Eline E C M Elsten,[1,11] Steffen Eychmüller,[8] Claudia Fischer ![ORCID],[6] Carl Johan Fürst,[12,13] Eric C T Geijteman,[1,11] Gabriel Goldraij,[14] Anne Goossensen,[15] Svandis Iris Halfdanardottir,[16] Dagny Faksvåg Haugen,[17,18] Christel Hedman,[12,13,19] Tanja Hoppe,[20] Rosemary Hughes,[10] Grethe Skorpen Iversen,[17] Melanie Joshi,[20] Hana Kodba-Ceh,[3] Ida J Korfage,[1] Urska Lunder,[3] Nora Lüthi,[8] Maria Luisa Martín-Roselló,[4,21] Stephen Mason,[10] Tamsin McGlinchey,[10] Silvi Montilla,[22] Birgit H Rasmussen,[12] Inmaculada Ruiz-Torreras,[4,21] Maria E C Schelin,[12,13] Katrin Ruth Sigurdardottir,[17,23] Valgerdur Sigurdardottir,[16] Judit Simon,[6] Ruthmarijke Smeding,[10] Kjersti Solvåg,[17] Julia Strupp ![ORCID],[20] Vilma Tripodoro,[9,22] Hugo M van der Kuy ![ORCID],[24] Carin C D van der Rijt,[11] Lia van Zuylen,[25] Verónica I Veloso,[22] Eva Vibora-Martin,[4] Raymond Voltz,[20,26,27,28] Sofia C Zambrano,[8,29] Agnes van der Heide[1]

For numbered affiliations see end of article.

**Correspondence to**
Berivan Yildiz;
b.yildiz@erasmusmc.nl

## ABSTRACT

**Introduction** Adequately addressing the needs of patients at the end of life and their relatives is pivotal in preventing unnecessary suffering and optimising their quality of life. The purpose of the iLIVE study is to contribute to high-quality personalised care at the end of life in different countries and cultures, by investigating the experiences, concerns, preferences and use of care of terminally ill patients and their families.

**Methods and analysis** The iLIVE study is an international cohort study in which patients with an estimated life expectancy of 6 months or less are followed up until they die. In total, 2200 patients will be included in 11 countries, that is, 200 per country. In addition, one relative per patient is invited to participate. All participants will be asked to fill in a questionnaire, at baseline and after 4 weeks. If a patient dies within 6 months of follow-up, the relative will be asked to fill in a post-bereavement questionnaire. Healthcare use in the last week of life will be evaluated as well; healthcare staff who attended the patient will be asked to fill in a brief questionnaire to evaluate the care that was provided. Qualitative interviews will be conducted with patients, relatives and healthcare professionals in all countries to gain more in-depth insights.

**Ethics and dissemination** The cohort study has been approved by ethics committees and the institutional review boards (IRBs) of participating institutes in all countries. Results will be disseminated through the project website, publications in scientific journals and at conferences. Within the project, there will be a working group focusing on enhancing the engagement of the community at large with the reality of death and dying.

**Trial registration number** NCT04271085.

## STRENGTHS AND LIMITATIONS OF THIS STUDY

⇒ Due to the international nature of this study, we are able to investigate end-of-life experiences across different cultures and among groups varying by age, gender, illness and care setting.

⇒ This study combines the perspectives of the most relevant stakeholders: patients who are in the last phase of life and their relatives, as well as healthcare staff providing end-of-life care.

⇒ The study population is relatively large which enables to perform subgroup analyses.

⇒ Although patients in the last phase of life and their caregivers have repeatedly reported to appreciate being given the opportunity to participate in research studies, completing a questionnaire about concerns, preferences and expectations concerning the end of life can be uncomfortable.

## INTRODUCTION

Over the past decades, increasing attention has been given to improving care for people

in the last phase of life. Literature suggests that most people wish to be free from pain and other symptoms, to be treated with dignity and respect, and to maintain a sense of autonomy and control over their last days.[1–4] In addition, many individuals wish to be informed of their limited life expectancy.[5] However, there is a substantial amount of variation in the definition of a 'good death'. Preferences for the end of life are dynamic and influenced by individual and multidimensional characteristics, such as age, gender, illness, care setting, financial resources, culture and social relationships.[6]

Medical care at the end of life is not optimally addressing the needs and preferences of all patients.[7] This is in many cases caused by barriers such as the unpredictable course of a terminal illness, communication difficulties and the complexity of care needs of dying patients and their families.[8] Many terminally ill patients are, for instance, unable to express their goals and preferences for medical treatment or care, due to physical deterioration or mental incapacity.[9 10] Moreover, since clinicians tend to focus on diagnosis, therapy and cure, the imminence of death is often not openly and timely acknowledged in patients with an advancing chronic illness.[11 12] A recent longitudinal study reported that end-of-life care was discussed between physicians and patients with terminal cancer in less than 20% of cases, and the frequency of these discussions only increased significantly in the last month of life.[13] Consequently, patients often receive treatment aimed at prolonging life until a very late stage in their illness trajectory, with a considerable burden for the patient.[14] Inadequately addressing the needs of the patients not only deteriorates the quality of patients' last phase of life[15] but also increases the risk of complicated grief in bereaved family members.[16]

So far, studies have mostly explored the perspectives and experiences regarding factors that are important in end-of-life care of citizens and physicians,[17–19] but the need to include the perspective of patients and their relatives has been acknowledged as well.[4 8 20] Studies investigating the needs and preferences of patients in their last phase of life have mostly included patients with cancer, and studied preferences on specific components of palliative care.[21] Little is known on patients' concerns, goals and sources of strength during their last phase of life.[22] In addition, no studies have investigated these aspects within a context of diversity in diagnosis, culture, gender and age.

We expect that patients in the last phase of life consider dignity, respect, social relations, autonomy, symptoms and pain control as important. Although some of these themes may be universal, we hypothesise that differences will exist in concerns, expectations and preferences based on gender, age, illness, care setting and culture.

The first aim of the iLIVE study is to provide in-depth understanding of the experiences, concerns, expectations and preferences of patients in the last phase of life and their relatives. The second aim is to assess variability in these concerns, expectations and preferences by culture, gender, and age, healthcare-related and socioeconomic factors. The international character of the iLIVE study provides a framework for unprecedented international comparative insights. A better understanding of needs and outcomes in end-of-life care will thus contribute to the development and advancement of policies to support dignified dying in various cultures and settings.

## METHODS AND ANALYSIS
### Study design and setting
The iLIVE study is a prospective observational cohort study involving terminally ill patients in hospital and non-hospital sites in 11 participating countries: Argentina, Germany, Iceland, the Netherlands, New Zealand, Norway, Slovenia, Sweden, Switzerland, Spain and the United Kingdom (UK). Countries from three continents over the world were included in the study to ensure cultural diversity within the study population. Terminally ill patients will be followed until they die or for a maximum of 6 months after inclusion. Participating patients and one of their relatives will complete questionnaires about their experiences, concerns, expectations and preferences around dying and use of end-of-life care. This 4-year study started in September 2020 and is currently ongoing.

In order to have a diverse study population regarding clinical and sociodemographic characteristics, we will recruit participants from different types of clinical settings. Patients will be recruited in the 11 participating countries, from a total of 20 hospitals (oncology, internal medicine, surgery, palliative care unit, medical physics, thoracic medicine and pulmonology departments), 7 specialised palliative care institutes and 8 out-of-hospital settings (nursing homes).

### Study population
In total, 2200 patients with a maximum estimated life expectancy of 6 months will be included, regardless of their diagnosis, gender or place of residence (table 1). Eligibility is assessed using a modified version of the Gold Standards Framework Proactive Identification Guidance (GSF-PIG) and the Supportive and Palliative Care Indicators Tool 2017 (SPICT).[20] The GSF-PIG starts with the 'surprise question', asking whether the physician would be surprised if a patient were to die within 1 year.[23] For the present study, we adapted this question into whether the physician would be surprised if a patient were to die within 6 months. If the physician is uncertain about the surprise question, the patient is eligible when at least one SPICT indicator is present.[24] SPICT is a tool to identify persons with poor or deteriorating health for assessment and care planning, using general indicators and clinical signs of life-limiting conditions (online supplemental table 1). All physicians will be informed on how to apply the GSF-PIG and the SPICT tool to assess eligibility.

Participating patients are asked to identify a relative, for instance, a family member or friend. Relatives are eligible

**Table 1** Inclusion and exclusion criteria for patients and relatives

| Inclusion criteria for patients | Exclusion criteria for patients and relatives |
|---|---|
| 18 years of age or older | Unable to provide informed consent |
| Attending physician would not be surprised if the patient were to die within 6 months | Incapable of filling in questionnaires in the country's main language or in English |
| In case of uncertainty about surprise question: at least one SPICT indicator | |
| Awareness that recovery is unlikely | |
| Written informed consent to participate | |
| **Inclusion criteria for relatives** | |
| 18 years of age or older | |
| Awareness that recovery of the patient is unlikely | |
| Written informed consent to participate | |
| SPICT, Supportive and Palliative Care Indicators Tool 2017. | |

if they are 18 years of age or older. Patients and relatives need to be aware that the patient is unlikely to recover from his or her illness. The exclusion criteria for patients also apply to relatives.

## Recruitment procedure

In the 11 countries, across all participating clinical sites, physicians are responsible for screening patients for eligibility. All consecutive patients admitted to a clinical ward or visiting an outpatient clinic will be screened for eligibility. Eligible patients are informed about the study by their attending physician or nurse, who provides them with an information leaflet. If patients agree to be informed about study participation, a researcher or research nurse from the local study team contacts them, answers their questions and asks them if they consent to participate. If the patient consents, the researcher asks them to consider whether a close relative might also be willing to participate. After obtaining written informed consent from patients and, if applicable, relatives, they will be asked to fill in the baseline questionnaire.

In each country, five patients, five relatives and five healthcare professionals will be interviewed. Patients and relatives completing the questionnaire face-to-face will be asked whether they are interested in an additional in-depth interview. Patients and relatives completing the questionnaire online or on paper (by post) will be approached by telephone. Patients and relatives who do not participate in the questionnaire study are also allowed to participate. If patients and/or relatives are eligible and

interested, the researcher or research nurse approaches them to explain further procedures and to conduct the interview. They will have the option of participating in a face-to-face or Skype interview.

Interviews will be conducted with healthcare professionals who are employed in the participating sites. Two criteria will be guiding the selection of healthcare professionals: (1) their work includes end-of-life care and (2) they have several years of experience with end-of-life care. There will be variation in profession and work setting among participants. The healthcare professional will be contacted by telephone or email inviting them to take part in the study.

## Measurements

The iLIVE cohort study includes several measurements (table 2):

1. Questionnaires. Patients, relatives and attending physicians are asked to fill in questionnaires. Patients and relatives will complete questionnaires on enrolment in the study (baseline assessment) and 4 weeks later (follow-up 1). For patients who die during the follow-up period of 6 months, relatives will also complete a questionnaire 8–10 weeks after the death of the patient (follow-up 2). Questionnaires for patients and relatives are administered on paper, online or through telephone or face-to-face interviews. Physicians will complete a paper questionnaire at patient enrolment (baseline assessment) and after the death of a patient (follow-up 2).

Completing the questionnaire will take approximately 30–45 min. In the online version of the questionnaire, participants are allowed to save their answers and continue at a later time point. The same is applicable to completing the paper version of the questionnaire and the face-to-face interview.

### Baseline assessment

The baseline questionnaire for patients includes questions on their experiences, concerns, expectations and preferences around dying and end-of-life care. Questions also address health-related quality of life, symptoms, decision-making, social support and attitudes towards euthanasia. Finally, questions are asked about health economic aspects, such as patients' employment status, use of healthcare and informal care needs. Relatives will also complete a questionnaire about their experiences, concerns, expectations and preferences around the last phase of life of the patient, their own health-related quality of life, their employment status and their provision of informal care. Attending physicians fill in a questionnaire about patients' diagnosis, comorbidities, life expectancy and their perspective on patients' current treatment aims. Where possible, validated measures that are commonly used to evaluate important aspects in end-of-life care are used to collect the data (table 2).

**Table 2** Measurements among patients, relatives and physicians within the iLIVE project

| I. Measured by questionnaire | Measurement instrument |
|---|---|
| **Patients** | |
| Concerns, expectations and preferences of patients around dying and end-of-life care | Self-developed questions adapted from the Serious Illness Conversation Guide[44] and the AEOLI questionnaire[45] |
| Symptom load | Edmonton Symptom Assessment System (ESAS)[46] |
| Health-related quality of life (HRQoL) and well-being | EORTC QLQ-C15-PAL quality of life question[47] and EuroQol 5 Dimension questionnaire (EQ-5D-5L)[48] ICECAP Supportive Care Measure (ICECAP-SCM)[49] |
| Attitudes towards euthanasia* | 10-item Euthanasia scale[50] |
| Health and social care resource use, absenteeism from work | (Partial) Health Economics Questionnaire (HEQ)[51] |
| Sociodemographic characteristics | Self-developed questions and HEQ |
| **Relatives** | |
| Concerns, expectations and preferences around dying and end-of-life care | Self-developed questions inspired by the Serious Illness Conversation Guide and the AEOLI questionnaire |
| Health-related quality of life (HRQoL) | EORTC QLQ-C15-PAL and EQ-5D-5L |
| Well-being | ICECAP Close Person Questionnaire (ICECAP-CPM)[52] |
| Informal care provision | iMTA Valuation of Informal Care Questionnaire (iVICQ)[53] and Informal Care Cost Assessment Questionnaire (CIIQ)[54] |
| Attitudes towards euthanasia | 10-item Euthanasia scale |
| Bereavement | Hogan Grief Reaction Checklist (HGRC, despair and personal growth subscales)[55] |
| Quality of care for dying patients | International questionnaire Care of the Dying Evaluation (iCODE)[56] |
| **Physicians** | |
| Patients' diagnosis, co-morbidities and life expectancy, perspective on patients' treatment aims and functional status | Based on the SPICT-criteria and the Australian version of the Karnofsky Performance Status[57] |
| Evaluation of care in the dying phase | Adapted and based on the Swedish Quality of Dying Registry[58] |
| **II. Obtained from medical files** | |
| Use of medical interventions, medication and costs of medical care in the last week of life | |
| Patient survival | |
| **III. Obtained from qualitative interviews** | |
| In-depth insights into experiences, concerns, expectations and preferences around dying and end-of-life care among patients, relatives and healthcare professionals | |

*In Norway and Iceland, one self-developed question will be used instead of the 10-item Euthanasia scale. No questions will be asked about euthanasia in Germany. Researchers from these countries were concerned that study participants would become anxious by these questions.

## Follow-up 1

Four weeks after the baseline assessment, patients and relatives are asked to complete a follow-up questionnaire to assess changes as compared with baseline.

## Follow-up 2

In case a participating patient dies, participating relatives are after 8–10 weeks asked to fill in a post-bereavement questionnaire, to assess their experience of the last days of life of the deceased patient, their appreciation of the quality of end-of-life care and family support, and their bereavement process. The physician or another healthcare staff member who attended the patient in the dying phase is also asked to complete a questionnaire to evaluate care in the dying phase. More specifically, questions will be asked on the place of death, symptoms and if they were treated, whether the patient and the family were informed that the patient was in the final stage of life, how long before death the patient lost the ability to express his/her will, and whether anyone was present at the time of death.

2. Medical file. Healthcare use in the patient's last week of life is assessed using a checklist. Items to be assessed include place of care, medical complications, medication use, major medical and surgical interventions and care, goals of care statements, resuscitation policy and non-treatment decisions.

3. Qualitative interviews. More in-depth insight will be obtained in complementary personal interviews with

patients, relatives and healthcare professionals. The same eligibility criteria apply as in the cohort study. The sample of interviewees will be controlled for age and gender per country, to allow a comparative analysis. The interviews will be semistructured using a topic guide that is based on Giger-Davidhizar-Haff's model for cultural assessment in end-of-life care,[25] the ABCD model for effectively addressing and integrating cultural needs and issues in clinical care,[26] and perception of disease questions.[27]

During the interviews with patients, questions will be asked about their understanding of the illness, relationship with family, concerns, difficulties to discuss end-of-life topics and decision-making. Comparable questions about these topics will be asked to relatives. Healthcare professionals will be asked questions about the care they aim to provide, collaboration with other professionals, communication with patients, decision-making, and values and beliefs when working with dying patients.

## Translation of questionnaires

Where possible, published and validated versions of existing instruments in the languages of the participating countries will be used. Where necessary, instruments will be translated. An instrument that has been translated correctly is conceptually equivalent to the source instrument[28–30] and thereby enables collection and pooling data from various linguistic and cultural regions. Translations will be performed according to the standard proposed by the EORTC Quality of Life Group.[31] The translation process will thus include two forward translations from English to the target language, development of a provisional consensus version, two backward translations and a careful comparison with the original. This will be repeated iteratively until a satisfactory result is obtained. The original developers of the instruments will provide feedback during this process and approve the final translations. Self-developed questions will be developed in English and translated following the same standards. The final translations will also be tested as part of the study questionnaire pilot testing in each country.

## Data management

This study will be conducted in accordance with the General Data Protection Regulation and national research ethics and privacy guidelines.[32] One common data management system will be used to safely process and store data of all participating patients, relatives and physicians across clinical sites and countries. In some countries, participants can choose to directly enter data into this system; in that case, they consent to use of their email address for communication purposes. In all other cases, data are entered anonymously by selected local research assistants, and a study number will be generated to link data of participants with a local communication database.

## Sample size

The primary outcomes are measured at baseline and 4 weeks post-inclusion. It is expected that 30% of all patients who complete the baseline assessment will be lost to follow-up, due to death, significant deterioration of health or other causes. In that case, 70% of patients who complete the baseline measurement will be able to complete the assessment after 4 weeks at follow-up 1. Further, it is expected that 80% of all patients who complete the baseline assessment can be followed until death, whereas the remaining 20% are expected to either survive until the end of the data collection period or become lost to follow-up. Regarding the relatives, it is expected that in case patients who complete the baseline assessment die during follow-up, half of the bereaved relatives (ie, 40% of all baseline patients), will be willing to complete a post-bereavement questionnaire (follow-up 2). The total cohort would thus include 2200 patients (n=200 per country) at baseline, 1540 patients (n=140 per country) at follow-up assessment 1 and 880 bereaved relatives (n=80 per country). The number of 200 patients per country enables us to estimate proportions with 95% Confidence Intervals (CIs) of approximately ±7%. The number of recruiting sites will vary from two to six per country.

No sample size estimation has been performed for the qualitative interviews since the aim is to explore and better understand the variety in experiences of patients, relatives and physicians, rather than having a representative sample per country.

## Analysis plan
### Primary outcomes

The primary outcomes are experiences, concerns, expectations and preferences around dying and end-of-life care of patients in the last phase of life and their relatives, at baseline and after 4 weeks follow-up, and will be described in frequencies and narrative descriptions. The proportion of patients who have certain concerns, expectations and preferences will be described. Subgroup analyses will be performed to assess cross-gender, cross-age and cross-cultural variety on experiences, concerns, expectations and preferences. Narrative descriptions will be translated into English and categorised into themes that will be identified within the data.

Descriptive statistics will be used to summarise baseline characteristics of the study participants (age, gender, education, diagnosis, comorbidities, religion, socioeconomic status, marital status, place of residence, quality of life, symptom load) by country and site. Statistics on mean/median scores and variance will be presented where applicable. Associations with country and patient characteristics will be analysed in a multilevel modelling approach, taking account of clustering effects at country level. Both univariable and multivariable analyses will be performed. All statistical tests will be two-sided and considered significant if $p<0.05$. Repeated measures analyses of variance will be conducted to assess the development

of outcomes between baseline and 4 weeks follow-up. Multivariate Imputation by Chained Equations (MICE) will be used to handle missing data,[33] as we expect that patients may not be able or want to fill in all questions in the questionnaire. MICE is known to be a flexible principled method of addressing missing data and can handle variables of varying types (eg, continuous or binary). Quantitative analyses will be performed with SPSS V.25.0 statistical software.

### Secondary outcomes

Secondary outcomes for patients include symptom load, health-related quality of life (HRQoL) and well-being, and attitudes towards physician assistance in dying. Secondary outcomes for relatives include HRQoL, well-being, informal care provision, attitudes towards physician assistance in dying and bereavement. The prevalence of these outcomes will be described in frequencies, mean/median scores and variance. Associations with country and patient characteristics will be analysed in a multilevel modelling approach, taking account of clustering effects at country level. Both univariable and multivariable analyses will be performed. Repeated measures analyses of variance will be conducted to assess the development of outcomes between baseline and 4 weeks follow-up. The relationship of the relative to the patient will be taken into account in multivariable models, in addition to the characteristics mentioned for the analysis of the primary outcome.

### Health economic analysis

The outcomes as assessed in this study allow inter alia for a comprehensive assessment of health resource utilisation and costs for medication and care, as well as patients' and relatives' quality of life and well-being. The study therefore includes a cost-effectiveness analyses of interventions used in end-of-life care. In addition, a framework for the value assessment of palliative and end-of-life care will be developed.[34]

### Qualitative interviews

The interviews will be recorded and transcribed verbatim. Data will be thematically analysed in an iterative process on different levels: within each country, within three subgroups of countries and across all countries. The analysis will be focused on identifying experiences, concerns, expectations and preferences, as well as underlying values and norms. In addition, comparison of patients' perspectives will be explored by gender and age to gain a better understanding of differences in phenomena between subgroups. Data from the interviews will be imported into NVivo software for analysis.

### Embedded intervention studies

The iLIVE project includes a number of studies that are embedded in the cohort study. The research protocols for these studies will be described elsewhere. A brief description is presented here.

### iLIVE medication study

Discussion of appropriate medication to alleviate symptoms is one of the key clinical issues in improving care for dying patients.[35] At the same time, potentially inappropriate medication is often continued until a very late stage in patients' illness trajectory.[36] This concern will be addressed in the iLIVE Medication Study, in which a digital clinical tool, a so-called Clinical Decision Support System (CDSS), will be used to optimise medication management in the last phase of life. A previous version of this tool to guide physicians in medication prescription and de-prescription for residents of nursing homes was developed and tested in the Netherlands.[37] In the iLIVE project, we developed an adapted version of this CDSS, the CDSS-OPTIMED, that supports physicians in optimising their prescription of medications for patients with a limited life expectancy. The CDSS-OPTIMED will be evaluated in three countries participating in the iLIVE project (the Netherlands, Sweden and Switzerland).

### iLIVE volunteer study

Volunteer services to support patients dying in hospitals, and their families, are relatively uncommon and empirical evidence of the usefulness of such services is scarce. This concern will be addressed in the iLIVE Volunteer Study, in which an international hospital palliative care volunteer training programme will be developed. This programme will underpin the implementation of palliative care volunteer services to support patients dying in hospital and their families, within five participating hospitals in five countries (the Netherlands, the UK, Norway, Slovenia and Spain). The iLIVE Volunteer Study will evaluate the implementation, use and experience of the iLIVE Volunteer Service.

### Core Outcome Set for care of the dying

It is important to identify the most important outcomes for care of dying patients through the perspective of patients, family members, researchers and health professionals. Despite a variety of available tools to assess different dimensions of palliative care, there is no consensus yet on which outcomes need to be measured in the last days of life. Therefore, this project will establish a Core Outcome Set (COS) for care of dying patients that includes valid, reliable and precise outcomes to enable international benchmarking, quality improvement and research in the last days of life.[38] In each country, patients and relatives will be invited to participate during this process.

### Patient and public involvement

An Advisory Board (AB) will be established with research and clinical experts and representatives from all relevant stakeholder groups: current and future patients and their families, healthcare professionals, volunteers, policy makers and researchers. The AB will engage in and advise on various aspects of the iLIVE project to ensure that the widest perspective on the process and outcomes can be realised.

In addition, in order to test the data collection for their acceptability and to maximise feasibility, we will pilot test the questionnaires in each country with three to five members from the target groups. Participants will be interviewed about their appreciation of the questionnaire following principles from cognitive interviewing techniques, which include open-ended questions as well specific probes (questions about potential problems). In case any modifications appear warranted, these will be discussed with the Project General Assembly. If relevant, modifications will be tested in additional patients.

## ETHICS AND DISSEMINATION

The study will be conducted in accordance with national and international regulations and guidelines, including the Declaration of Helsinki,[39] and the International Conference on Harmonisation (ICH) guidance on Good Clinical Practice (GCP).[40] The study has been approved by ethics committees and institutional review boards (IRBs) in all participating countries. The following ethics committees have approved the study:

► Regional Committee for Medical and Health Research Ethics South East D (35035), Norway.
► Komisija Republike Slovenije za Medicinsko etiko (0120-129/2020/3), Slovenia.
► Health Research Authority (HRA) and Health and Care Research Wales (HCRW) (272927), UK.
► Comité de Ética de la Investigación Provincial de Málaga, Hospital Regional Universitario de Malaga, Spain.
► Swedish Ethical Review Authority (2020-01956), Lund University, Sweden.
► The National Bioethics Committee (VSN-20-129), Iceland.
► Ethics Commission of Cologne University, Faculty of Medicine (19-1456_1).
► Gesundheits-, Sozial und Integrationdirektion Kantonale Ethikkommission fur die Forschung (2020-02569), Switzerland.
► Medical Research Ethics Committees United (MEC-U) (R20.004), The Netherlands.
► Dictamen del Comité de ética del instituto Lanari, University of Buenos Aires.

This study is registered in ClinicalTrials.gov. A Data Safety Monitoring Board has been established.

All potential participants to the study are provided with oral and written information about the study in the country's language. They will be given at least 72 hours (3 days) to consider participation and ask questions. All participants will be asked to provide written informed consent to confirm their willingness to participate in the study and for the data collection, storage and transfer of data according to established procedures.

We acknowledge the potential vulnerability of patients in the last phase of life and their relatives, and the risk of overburdening. Completing a questionnaire about concerns, preferences and expectations concerning the end of life can be uncomfortable. However, patients in the last phase of life and their caregivers have repeatedly been reported to appreciate being given the opportunity to participate in research studies, even when they are close to death.[41 42] Participation in this study may nevertheless cause emotional burden for patients. Study participants will as a matter of principle be approached as people who are in principle fully capable of participating in research and whose experiences and concerns are important for healthcare professionals to learn from. If patients feel burdened by their participation, they are encouraged to indicate that on the questionnaire or to the researcher. Patients are also encouraged to discuss their issues with relatives or a healthcare professional.

The project results will be disseminated through the project website (www.iliveproject.eu), publications in scientific journals and at conferences. Within the project, there will be a working group focusing on enhancing the engagement of the community at large with the reality of death and dying. One of the aims is to actively promote societal debate and engagement with death and dying. This will be achieved by developing a detailed dissemination plan for efficient engagement of citizens, patients and families, healthcare professionals, volunteers and policy makers throughout the project, and effective dissemination of emerging outcomes.

## DISCUSSION

The iLIVE study has several strengths. Going through the last phase of life is a complex personal experience, which is best understood while acknowledging the diverse and dynamic preferences of patients and their families. Due to the international nature of this project, we are able to investigate end-of-life experiences across different cultures and among groups varying by age, gender, illness and care setting. Further, we combine the perspectives of the most relevant stakeholders, that is, patients who are in the last phase of life and their relatives, as well as healthcare staff providing end-of-life care. Furthermore, patients and relatives will complete questionnaires at multiple time points, which enables us to analyse potential adaptations within subjects over time. Another strength relates to the post-bereavement assessment among relatives, which provides insights in the experience of care in the dying phase as well as the impact of these experiences on relatives' well-being and bereavement after the death of a patient. Lastly, the study population is relatively large which enables us to perform subgroup analyses.

We expect to encounter several challenges in this study. Recruiting patients in the last phase of life for research studies is often difficult. For instance, healthcare professionals or family members may be hesitant to provide researchers' access to incurably ill patients, due to concerns about burdening or distressing them, a phenomenon referred to as 'gatekeeping'.[43] In many studies, this has led to considerably smaller study samples than desired. To minimise this risk, we have involved

multiple clinical sites in almost all participating countries, planned for modest numbers of participants per site and applied conservative estimates of expected dropout. In addition, we will screen all potentially eligible patients and keep track of inclusion and exclusion numbers, as well as reasons for non-participation or exclusion.

Another challenge is that persons at the end of life may not be able to complete the follow-up questionnaire as they may become weaker over time. This will be monitored during the study and necessary actions will be taken in order to improve completion of the follow-up questionnaire.

In conclusion, the iLIVE project is aimed at increasing our understanding of the experience of dying in different settings and cultures around the world, and of the concerns, expectations, preferences and needs of dying patients and their relatives. Such understanding is currently lacking, but key to the development of effective and efficient palliative and end-of-life care and public health policies.

**Author affiliations**
[1]Department of Public Health, Erasmus MC, University Medical Center Rotterdam, Rotterdam, The Netherlands
[2]Arohanui Hospice, Palmerston North, New Zealand
[3]Research Department, University Clinic of Respiratory and Allergic Diseases Golnik, Golnik, Slovenia
[4]CUDECA Institute for Training and Research in Palliative Care, CUDECA Hospice Foundation, Malaga, Spain
[5]Group C08: Pharma Economy: Clinical and Economic Evaluation of Medication and Palliative Care, Ibima Institute, Malaga, Spain
[6]Department of Health Economics, Center for Public Health, Medical University of Vienna, Wien, Austria
[7]Department of Palliative Care, St Vincent's Hospital Melbourne, Fitzroy, Victoria, Australia
[8]University Center for Palliative Care, Inselspital University Hospital Bern, University of Bern, Bern, Switzerland
[9]Research Network RED-InPal, Institute Pallium Latinoamérica, Buenos Aires, Argentina
[10]Palliative Care Unit, Institute of Life Course and Medical Sciences, University of Liverpool, Liverpool, UK
[11]Department of Medical Oncology, Erasmus MC Cancer Institute, Erasmus MC University Medical Center Rotterdam, Rotterdam, The Netherlands
[12]Institute for Palliative Care at Lund University and Region Skåne, Lund University, Lund, Sweden
[13]Division of Oncology and Pathology, Department of Clinical Sciences, Lund University, Lund, Sweden
[14]Internal Medicine/Palliative Care Program, Hospital Privado Universitario de Córdoba, Cordoba, Argentina
[15]Informal Care and Care Ethics, University of Humanistic Studies, Utrecht, The Netherlands
[16]Palliative Care Unit, Landspitali - National University Hospital, Reykjavik, Iceland
[17]Regional Centre of Excellence for Palliative Care, Haukeland University Hospital, Western Norway, Bergen, Norway
[18]Department of Clinical Medicine K1, University of Bergen, Bergen, Norway
[19]Research Department, Stiftelsen Stockholms Sjukhem, Stockholm, Sweden
[20]Department of Palliative Medicine, Faculty of Medicine and University Hospital, University of Cologne, Cologne, Germany
[21]Group CA15: Palliative Care, IBIMA Institute, Malaga, Spain
[22]Institute of Medical Research A. Lanari, University of Buenos Aires, Buenos Aires, Argentina
[23]Specialist Palliative Care Team, Department of Anaesthesia and Surgical Services, Haukeland University Hospital, Bergen, Norway
[24]Department of Clinical Pharmacy, Erasmus MC, University Medical Center, Rotterdam, The Netherlands
[25]Department of Medical Oncology, Amsterdam University Medical Center, Amsterdam, The Netherlands
[26]Center for Integrated Oncology Aachen Bonn Cologne Dusseldorf (CIO ABCD), Faculty of Medicine and University Hospital, University of Cologne, Cologne, Germany
[27]Clinical Trials Center (ZKS), Faculty of Medicine and University Hospital, University of Cologne, Cologne, Germany
[28]Center for Health Services Research (ZVFK), Faculty of Medicine and University Hospital, Cologne, Germany
[29]Institute for Social and Preventive Medicine (ISPM), University of Bern, Bern, Switzerland

**Correction notice** This article has been corrected since it first published. Author name 'Judit Simon' has been updated.

**Contributors** The consortium has evolved from a successful collaboration of clinicians and researchers that developed through a joint interest and expertise in care of the dying to the 'International Collaborative for Best Care for the Dying Person'. AVDH, IJD, SA, MB7, ME, JE, SE, CF, CJF, EG, AG, SIH, DFH, CH, TH, UL, MLMR, SM, BHR, MS, KRS, VS, JS, RM, KS, JS, VAT, HVDK, CCDVDR, LVZ, RV, and SCZ contributed to the design of the study. BY wrote the first draft of the manuscript and revised with input from all authors. AVDH, IJD, SA, MB7, ME, JE, SE, CF, CJF, EG, AG, SIH, DFH, CH, TH, UL, MLMR, SM, BHR, MS, KRS, VS, JS, RM, KS, JS, VAT, HVDK, CCDVDR, LVZ, RV, SCZ, MB3, PBF, MB6, AC, EECME, RH, GSI, MJ, HKC, NL, TMG and IRT critically reviewed the manuscript for important intellectual content. GGDS, GG, SM, VIV, EVM, MB3, PBF, EECME, SIH, MJ, HKC, TMG, BHR, IRT, KS, VAT, EVB and SCZ provided feedback on the manuscript. All authors read and approved the final manuscript.

**Funding** This work is supported by the European Union's Horizon 2020 Research and Innovation Programme under Grant agreement no. 825731.

**Competing interests** None declared.

**Patient and public involvement** Patients and/or the public were not involved in the design, or conduct, or reporting or dissemination plans of this research.

**Patient consent for publication** Not applicable.

**Provenance and peer review** Not commissioned; externally peer reviewed.

**ORCID iDs**
Berivan Yildiz http://orcid.org/0000-0002-2385-213X
Claudia Fischer http://orcid.org/0000-0001-7574-8097
Julia Strupp http://orcid.org/0000-0003-3135-2693
Hugo M van der Kuy http://orcid.org/0000-0002-7128-8801

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
