## [Reviewer comments · BMJ Open]

ARTICLE DETAILS

TITLE (PROVISIONAL)	LIVE WELL, DIE WELL - AN INTERNATIONAL COHORT STUDY ON EXPERIENCES, CONCERNS AND PREFERENCES OF PATIENTS IN THE LAST PHASE OF LIFE: THE RESEARCH PROTOCOL OF THE iLIVE STUDY
AUTHORS	Yildiz, Berivan; Allan, Simon; Bakan, Misa; Barnestein-Fonseca, Pilar; Berger, Michael; Boughey, Mark; Christen, Andri; De Simone, Gustavo; Egloff, Martina; Ellershaw, John; Elsten, Eline; Eychmüller, S; Fischer, Claudia; Fürst, Carl; Geijteman, Eric C.T.; Goldraij, Gabriel; Goossensen, Anne; Halfdanardottir, Svandis Iris; Faksvåg Haugen, Dagny; Hedman, Christel; Hoppe, Tanja; Hughes, Rosemary; Iversen, Grethe; Joshi, Melanie; Kodba-Ceh, Hana; Korfage, Ida; Lunder, Urska; Lüthi, Nora; Martín-Roselló, Maria; Mason, Stephen; MC Glinchey, Tamsin; Montilla, Silvi; Rasmussen, Birgit; Ruiz-Torres, Inmaculada; Schelin, Maria; Sigurdardottir, Katrin; Sigurdardottir, Valgerdur; Simon, Judith; Smeding, Ruthmarijke; Solvag, Kjersti; Strupp, Julia; Tripodoro, Vilma; van der Kuy, Hugo; van der Rijt, Carin; van Zuylen, Lia; Veloso, Verónica; Vibora-Martin, Eva; Voltz, Raymond; Zambrano, Sofia; van der Heide, Agnes

VERSION 1 – REVIEW

REVIEWER	Dominic Wilkinson University of Oxford, Oxford Uehiro Centre for Practical Ethics
REVIEW RETURNED	23-Nov-2021

GENERAL COMMENTS	This is a fascinating and well planned large scale study that has the potential to provide real insights into the preferences and experiences of dying patients in 11 different countries. I had only a couple of small suggestions/comments. 1. Patient and Public Involvement This study is by no means unique in thinking about PPI largely in terms of dissemination of results, rather than in the design of the research itself. However, this does seem a missed opportunity - to involve patient groups in the different countries in providing input into the design of the study itself. I appreciate at this stage that may no longer be relevant, but it may be useful to think about ways of involving patients in the development of study materials and even potentially in interpretation of study results. 2. Information about institutions/recruitment sites The protocol provides very little information about the different recruitment sites in the 11 countries. How were they selected? Were there efforts to have centres of different types/size? Will differences in the source of patient between countries affect the interpretation of
---

	results? 3. Language What is the process for translation of materials and verification of translation between the different countries? 4. End of life decisions An evaluation of care (reported by physicians) is planned, but does this include any description or analysis of end of life decisions made? It would be helpful to clarify
--	---

REVIEWER	Hsiao-Ting Chang Taipei Veterans General Hospital, Department of Family Medicine
REVIEW RETURNED	01-Feb-2022

GENERAL COMMENTS	Reviewer's comments Manuscript Title: LIVE WELL, DIE WELL- AN INTERNATIONAL COHORT STUDY ON EXPERIENCES, CONCERNS AND PREFERENCES OF PATIENTS IN THE LAST PHASE OF LIFE: THE RESEARCH PROTOCOL OF THE iLIVE STUDY Reviewer's report: This study protocol aims to investigate the experiences, concerns, preferences and use of care of terminally ill patients and their families by questionnaire survey across 11 countries. This is an interesting study and the authors reported several significant issues in the study protocol. However, I have some recommendations and questions for authors as follows: Introduction: 1. Please add references related to the study outcomes from previous studies in this section and discuss potential gaps this study may fill. 2. What are the hypotheses in this study? Methods: Please add what date will the study be started or what date the study has been started and it is ongoing. Study design and setting: 1. Please explain the reasons to conduct this study in these countries. 2. Please describe the characteristics of study settings in each country. What are the non-hospital settings? What wards and what physician specialties will be responsible for patient screening? What about their experiences in clinical care and the use of GSF-PIG and SPICT? Study population 1. What sampling method will be used? 2. Are physicians also the participants in this study? How many of them will be recruited? And what are the inclusion and exclusion criteria for physicians? 3. How many participants will be recruited from each hospital setting and non-hospital setting, respectively? Recruitment procedure 1. "If patients are interested in participating, the researcher contacts them," What is the researcher mean? Is the researcher the physician or nurse or a research assistant? Measurements 1. The authors adapted many questionnaires in this study. Please
---

	explain the reasons to use these questionnaires and the validity and reliability for these questionnaires in different countries or different languages. 2. Questionnaires for patients and relatives are administered on paper, online, or through telephone or face-to-face interviews. The third part of measurements is “qualitative interviews”. Please explained how to select participants and how to conduct the qualitative interviews for participants who answer the questionnaires in different ways? Furthermore, please explain the sample size estimation for qualitative interview. 3. Please explain how to score each measurement instrument and the range of scores and the meanings of scores. 4. Please specify what version of SPICT will be used. 5. What are the outlines of the qualitative interview? 6. Please add descriptions for the ABCD model. 7. Please explain for “In Norway and Iceland, one self-developed question will be used instead of the 10-item Euthanasia scale. No questions will be asked about euthanasia in Germany.” 8. There are many measurement instruments, what is the expected time to complete all these measurements for each participant? If the participant is not able to complete all the measurement at one time, are there any alternative methods could help? If a patient is in the terminal stage, he/she may become weaker day by day and the follow-up time of 4 weeks may not be applicable to some of these patients. Sample size 1. Please explain how the total sample size was estimated and how many participants will be recruited in each setting. Analyses 1. Please cite references for “Multivariate Imputation by Chained Equations (MICE).....” and please explain the reasons. 2. What statistical software will be used to analyze quantitative and qualitative data, respectively? 3. Please add detailed descriptions for the descriptive and inferential statistics for the primary and secondary and other outcomes.
--	---

VERSION 1 – AUTHOR RESPONSE

Reviewer: 1

Dr. Dominic Wilkinson, University of Oxford

Comments to the Author:

This is a fascinating and well planned large scale study that has the potential to provide real insights into the preferences and experiences of dying patients in 11 different countries.

I had only a couple of small suggestions/comments.

General response to the comment the reviewer:

We thank the reviewer for acknowledging the relevance of this study. Below we will respond to each comment.

1. Patient and Public Involvement

This study is by no means unique in thinking about PPI largely in terms of dissemination of results, rather than in the design of the research itself. However, this does seem a missed opportunity - to involve patient groups in the different countries in providing input into the design of the study itself. I appreciate at this stage that may no longer be relevant, but it may be useful to think about ways of

involving patients in the development of study materials and even potentially in interpretation of study results.

Response to comment:

We acknowledge that patients groups have not been involved in the design of the study. However, we have conducted pilot tests for the questionnaires among patients. In addition, we will involve patients and relatives from each country to identify the Core Outcome set for care of the dying. Regarding the interpretation of the study results we will involve an advisory board to the project. These points are all addressed in the methods section:

New text (Methods, p. 15, line 9):

An Advisory Board (AB) will be established with research and clinical experts and representatives from all relevant stakeholder groups: current and future patients and their families, health care professionals, volunteers, policy makers, and researchers. The AB will engage in and advise on various aspects of the iLIVE project to ensure that the widest perspective on the process and outcomes can be realized.

In addition, in order to test the data collection for their acceptability and to maximize feasibility, we will pilot test the questionnaires in each country with 3-5 members from the target groups. Participants will be interviewed about their appreciation of the questionnaire following principles from cognitive interviewing techniques, which include open-ended questions as well specific probes (questions about potential problems). In case any modifications appear warranted, these will be discussed with the Project General Assembly. If relevant, modifications will be tested in additional patients.

New text (Methods, p. 15, line 1):

It is important to identify the most important outcomes for care of dying patients through the perspective of patients, family members, researchers, and health professionals. Despite a variety of available tools to assess different dimensions of palliative care, there is no consensus yet on which outcomes need to be measured in the last days of life. Therefore, this project will establish a Core Outcome Set (COS) for care of dying patients that includes valid, reliable and precise outcomes to enable international benchmarking, quality improvement and research in the last days of life. *In each country, patients and relatives will be invited to participate during this process.*

2. Information about institutions/recruitment sites

The protocol provides very little information about the different recruitment sites in the 11 countries. How were they selected? Were there efforts to have centres of different types/size? Will differences in the source of patient between countries affect the interpretation of results?

Response to the comment:

We aim to include participants from different types of clinical settings in order to have a study population with diversity in clinical and sociodemographic characteristics.

New text (Methods, p. 6, line 12):

In order to have a diverse study population regarding clinical and sociodemographic characteristics, we will recruit participants from different types of clinical settings. Patients will be recruited in the 11 participating countries, from in total 20 hospitals (oncology, internal medicine, surgery, palliative care unit, medical physics, thoracic medicine and pulmonology departments), 7 specialized palliative care institutes, and 8 out-of-hospital settings (nursing homes).

New text (Methods, p. 6, line 28):

All physicians will be informed on how to apply the GSF-PIG and the SPICT tool to assess eligibility.

3. Language

What is the process for translation of materials and verification of translation between the different countries?

Response to comment:

We have added a paragraph to explain the process of translation.

New text (Methods, p. 11, line 5):

Translation of questionnaires

Where possible, published and validated versions of existing instruments in the languages of the participating countries will be used. Where necessary, instruments will be translated. An instrument that has been translated correctly is conceptually equivalent to the source instrument (1-3) and thereby enables collection and pooling data from various linguistic and cultural regions. Translations will be performed according to the standard proposed by the EORTC Quality of Life Group (4). The translation process will thus include two forward translations from English to the target language, development of a provisional consensus version, two backward translations, and a careful comparison with the original. This will be repeated iteratively until a satisfactory result is obtained. The original developers of the instruments will provide feedback during this process and approve the final translations. Self-developed questions will be developed in English and translated following the same standards. The final translations will also be tested as part of the study questionnaire pilot testing in each country.

4. End of life decisions

An evaluation of care (reported by physicians) is planned, but does this include any description or analysis of end of life decisions made? It would be helpful to clarify

Response to comment:

The evaluation of care reported by physicians mainly includes the circumstances around the death of the participant rather than end-of-life decisions. We have added a clarification to the paragraph regarding the evaluation of care by physicians:

New text (Methods, p. 9, line 19):

More specifically, questions will be asked on the place of death, symptoms and if they were treated, whether the patient and the family were informed that the patient was in the final stage of life, how long before death the patient lost the ability to express his/her will, and whether anyone was present at the time of death.

Reviewer: 2

Dr. Hsiao-Ting Chang, Taipei Veterans General Hospital

Comments to the Author:

Reviewer's comments

Manuscript Title: LIVE WELL, DIE WELL- AN INTERNATIONAL COHORT STUDY ON EXPERIENCES, CONCERNS AND PREFERENCES OF PATIENTS IN THE LAST PHASE OF LIFE: THE RESEARCH PROTOCOL OF THE iLIVE STUDY

Reviewer's report:

This study protocol aims to investigate the experiences, concerns, preferences and use of care of terminally ill patients and their families by questionnaire survey across 11 countries. This is an interesting study and the authors reported several significant issues in the study protocol. However, I have some recommendations and questions for authors as follows:

Response to the comment:

We thank the reviewer for the compliments and the critical recommendations. Below we have answered all recommendations and questions.

Introduction:

1. Please add references related to the study outcomes from previous studies in this section and discuss potential gaps this study may fill.

Response to the comment:

We have added references to the first (5, 6) and second (7) paragraph of the introduction on recent studies on preferences regarding end-of-life care. We have added references to the last paragraph of the introduction and discussed the gaps in the literature.

Old text:

So far, studies have mostly explored the perspectives and experiences of communities and physicians regarding factors that are important in end-of-life care (8-10), but the need to include the perspective of patients and their relatives has been acknowledged as well (11, 12). Currently, however, there is a lack of knowledge on what patients in the last phase of life and their relatives consider important (13). The first aim of the iLIVE study is therefore to provide in-depth understanding of the experiences, concerns, expectations and preferences of patients in the last phase of life and their relatives. The second aim is to assess variability in these concerns, expectations and preferences by culture, gender, age, healthcare-related and socio-economic factors. The international character of the iLIVE study provides a framework for unprecedented international comparative insights. A better understanding of needs and outcomes in end-of-life care will thus contribute to the development and advancement of policies to support dignified dying in various cultures and settings.

New text (Introduction, p. 4, line 22):

So far, studies have mostly explored the perspectives and experiences regarding factors that are important in end-of-life care *of citizens and physicians* (8-10), but the need to include the perspective of patients and their relatives has been acknowledged as well (11, 12). *Studies investigating the needs and preferences of patients in their last phase of life have mostly included patients with cancer, and studied preferences on specific components of palliative care (14). Little is known on patients' concerns, goals and sources of strength during their last phase of life. In addition, no studies have investigated these aspects within a context of diversity in diagnosis, culture, gender and age.*

The first aim of the iLIVE study is to provide in-depth understanding of the experiences, concerns, expectations and preferences of patients in the last phase of life and their relatives. The second aim is to assess variability in these concerns, expectations and preferences by culture, gender, age, healthcare-related and socio-economic factors. The international character of the iLIVE study provides a framework for unprecedented international comparative insights. A better understanding of needs and outcomes in end-of-life care will thus contribute to the development and advancement of policies to support dignified dying in various cultures and settings.

2. What are the hypotheses in this study?

Response to the comment:

We have added a paragraph to the introduction about our hypothesis.

New text (Introduction, p. 4, line 30):

We expect that patients in the last phase of life consider dignity, respect, social relations, autonomy, symptoms and pain control as important. Although some of these themes may be universal, we hypothesise that differences will exist in concerns, expectations and preferences based on gender, age, illness, care setting and culture.

Methods:

Please add what date will the study be started or what date the study has been started and it is ongoing.

Response to the comment:

New text (Methods, p. 6, line 9):

This 4-year study started in September 2020 and is currently ongoing.

Study design and setting:**1. Please explain the reasons to conduct this study in these countries.**

New text (Methods, p. 6, line 5):

Countries from three continents over the world were included in the study to ensure cultural diversity within the study population.

2. Please describe the characteristics of study settings in each country. What are the non-hospital settings? What wards and what physician specialties will be responsible for patient screening? What about their experiences in clinical care and the use of GSF-PIG and SPICT?

Response to the comment:

We have addressed these comments in a new paragraph in the methods section (please see also our response to comment 2 of the first reviewer). We have unfortunately no insights into experiences of all physicians in clinical care and the use of GSF-PIG and SPICT.

New text (Methods, p. 6, line 12):

In order to have a diverse study population regarding clinical and sociodemographic characteristics, we will recruit participants from different types of clinical settings. Patients will be recruited in the 11 participating countries, from in total 20 hospitals (oncology, internal medicine, surgery, palliative care unit, medical physics, thoracic medicine and pulmonology departments), 7 specialized palliative care institutes, 8 out-of-hospital settings (nursing homes).

New text (Methods, p. 6, line 28):

All physicians will be informed on how to apply the GSF-PIG and the SPICT tool to assess eligibility.

Study population

1. What sampling method will be used?

Response to the comment:

We have added a sentence about the sampling method to the methods section:

New text (Methods, p. 7, line 9):

All consecutive patients admitted to a clinical ward or visiting an outpatient clinic will be screened for eligibility.

2. Are physicians also the participants in this study? How many of them will be recruited? And what are the inclusion and exclusion criteria for physicians?

Response to the comment:

In the questionnaire study, physicians only provide information on clinical characteristics of participants. However, in the qualitative interview study, physicians are considered as participants as they take part in a personal in-depth interview. The latter is addressed in the methods:

New text (Methods, p. 8, line 6):

Interviews will be conducted with health care professionals who are employed in the participating sites. Two criteria will be guiding the selection of healthcare professionals: (1) their work includes end-of-life care, and (2) they have several years of experience with end-of-life care. There will be variation in profession and work setting among participants. The healthcare professional will be contacted by telephone or email inviting them to take part in the study.

3. How many participants will be recruited from each hospital setting and non-hospital setting, respectively?

Response to the comment:

The power calculation concerned the number of participants from each country, i.e. 200 participants. There are no guidelines on how many participants should be recruited per clinical setting in each country. Therefore, there may be differences in the number of participants between hospital settings and non-hospital settings among the countries.

Recruitment procedure

1. "If patients are interested in participating, the researcher contacts them," What is the researcher mean? Is the researcher the physician or nurse or a research assistant?

Response to the comment:

We have clarified this in the methods section:

Old text:

If patients are interested in participating, the researcher contacts them, answers their questions, and asks them if they consent to participate.

New text (Methods, p. 7, line 11):

If patients *agree to be informed about study participation*, a researcher or research nurse from the *local study team* contacts them, answers their questions, and asks them if they consent to participate.

Measurements

1. The authors adapted many questionnaires in this study. Please explain the reasons to use these questionnaires and the validity and reliability for these questionnaires in different countries or different languages.

Response to the comment:

Thank you for this comment. We have clarified why we will use these questionnaires. If available, we will use validated instruments in the language of the participating countries. Instruments that are not available in the language of the countries will be translated. We have explained this process in the Methods:

Old text:

Where possible, validated measures are used to collect these data.

New text (Methods, p. 9, line 6):

Where possible, validated measures *that are commonly used to evaluate important aspects in end-of-life care* are used to collect the data.

New text (Methods, p. 11, line 5):

Translation of questionnaires

Where possible, published and validated versions of existing instruments in the languages of the participating countries will be used. Where necessary, instruments will be translated. An instrument that has been translated correctly is conceptually equivalent to the source instrument (1-3) and thereby enables collection and pooling data from various linguistic and cultural regions. Translations will be performed according to the standard proposed by the EORTC Quality of Life Group (4). The translation process will thus include two forward translations from English to the target language, development of a provisional consensus version, two backward translations, and a careful comparison with the original. This will be repeated iteratively until a satisfactory result is obtained. The original developers of the instruments will provide feedback during this process and approve the final translations. Self-developed questions will be developed in English and translated following the same standards. The final translations will also be tested as part of the study questionnaire pilot testing in each country.

2. Questionnaires for patients and relatives are administered on paper, online, or through telephone or face-to-face interviews. The third part of measurements is “qualitative interviews”. Please explained how to select participants and how to conduct the qualitative interviews for participants who answer the questionnaires in different ways? Furthermore, please explain the sample size estimation for qualitative interview.

Response to the comment:

For patients and relatives participating in the qualitative interviews, the same criteria are used as for the questionnaire study. We have added more details on how they and the physicians are selected:

New text (Methods, p. 7, line 17):

In each country, five patients, five relatives and five healthcare professionals will be interviewed. Patients and relatives completing the questionnaire face-to-face will be asked whether they are interested in an additional in-depth interview. Patients and relatives completing the questionnaire online or on paper (by post) will be approached by telephone. Patients and relatives who do not participate in the questionnaire study are also allowed to participate. In this case, the physician is responsible for assessing eligibility and recruiting participants. If patients and/or relatives are eligible and interested, the researcher or research nurse approaches them to explain further procedures and to conduct the interview. They will have the option of participating in a face-to-face or skype interview.

New text (Methods, p. 12, line 17):

No sample size estimation has been performed for the qualitative interviews since the aim is to explore and better understand the variety in experiences of patients, relatives, and physicians, rather than having a representative sample per country.

3. Please explain how to score each measurement instrument and the range of scores and the meanings of scores.

Response to the comment:

We have included a supplementary table in which we have included details on how to score each measurement instrument with its range and meaning:

Topic	Measurement instrument	Scale scores
Patients		
- Concerns, expectations and preferences of patients around dying and end-of-life care	- Self-developed questions adapted from the Serious Illness Conversation Guide (15) - AEOLI questionnaire (16)	Not applicable Strongly disagree – disagree- neither agree nor disagree – agree- strongly agree – don’t know
- Symptom load	Edmonton Symptom Assessment System (ESAS) (17)	0 (no symptom) – 10 (worst possible symptom)
- Health-related quality of life (HRQoL) and wellbeing	EORTC QLQ-C15-PAL quality of life question (18) EuroQol 5 Dimension questionnaire (EQ-5D-5L) (19)	0 (worst health) – 100 (best health) Questions 1-3: no problems – slight problems- moderate problems – severe problems - unable Questions 4 (pain) & 5 (anxious): no(t) – slight – moderate – severe – extreme(ly) Most of the time –some of the time – only a little of

	ICECAP Supportive Care Measure (ICECAP-SCM) (20)	the time - never
- Attitudes towards euthanasia ^a	10-item Euthanasia scale (21)	Strongly disagree – disagree- neither agree nor disagree – agree- strongly agree – don't know
- Health and social care resource use, absenteeism from work	(Partial) Health Economics Questionnaire (HEQ)(22)	Not applicable
- Sociodemographic characteristics	Self-developed questions and HEQ	Not applicable
Relatives		
- Concerns, expectations and preferences around dying and end-of-life care	Self-developed questions inspired by the Serious Illness Conversation Guide and the AEOLI questionnaire	Not applicable
- Health-related quality of life (HRQoL)	EORTC QLQ-C15-PAL EQ-5D-5L	0 (worst health) – 100 (best health) Questions 1-3: no problems – slight problems- moderate problems – severe problems - unable Questions 4 (pain) & 5 (anxious): no(t) – slight – moderate – severe – extreme(ly)
- Well-being	ICECAP Close Person Questionnaire (ICECAP-CPM) (23)	Question 1 -2 : all of the time- most- some- a little- non Question 3-6: fully able – mostly able- mostly unable –completely unable
- Informal care provision	iMTA Valuation of Informal Care Questionnaire (iVICQ)(24) and Informal Care Cost Assessment	Not applicable

	Questionnaire (CIIQ) (25)	
- Attitudes towards euthanasia	10-item Euthanasia scale	Strongly disagree – disagree- neither agree nor disagree – agree- strongly agree – don't know
- Bereavement	Hogan Grief Reaction Checklist (HGRC, despair and personal growth subscales) (26)	1= Does not describe me at all 2 = Does not quite describe me 3 = Describes me fairly well 4 = Describes me well 5 = Describes me very well
- Quality of care for dying patients	International questionnaire Care of the Dying Evaluation (iCODE) (27)	Various scales
Physicians		
- Patients' diagnosis, co-morbidities and life expectancy, perspective on patients' treatment aims and functional status	Based on the SPICT-criteria and the Australian version of the Karnofsky Performance Status (28)	Not applicable
- Evaluation of care in the dying phase	Adapted and based on the Swedish Quality of Dying Registry (29)	Various scales

4. Please specify what version of SPICT will be used.

Response to the comment:

New text (Methods, p. 6, line 20):

Eligibility is assessed using a modified version of the Gold Standards Framework Proactive Identification Guidance (GSF-PIG) and the Supportive and Palliative Care Indicators Tool 2017 (SPICT).

5. What are the outlines of the qualitative interview?

Response to the comment:

We have added details on the outlines of the qualitative interviews.

New text (Methods, p. 10, line 5):

During the interviews with patients, questions will be asked about their understanding of the illness, relationship with family, concerns, difficulties to discuss end-of-life topics, and decision-making. Comparable questions about these topics will be asked to relatives. Healthcare professionals will be asked questions about the care they aim to provide, collaboration with other professionals, communication with patients, decision-making, and values and beliefs when working with dying patients.

6. Please add descriptions for the ABCD model.

Response to the comment:

We have added a description for the ABCD model:

New text (Methods, p. 9, line 32):

The interviews will be semi-structured using a topic guide that is based on Giger-Davidhizar-Haff's model for cultural assessment in end-of-life care, the ABCD *model for effectively addressing and integrating cultural needs and issues in clinical care*, and perception of disease questions.

7. Please explain for "In Norway and Iceland, one self-developed question will be used instead of the 10-item Euthanasia scale. No questions will be asked about euthanasia in Germany."

New text (Methods, p. 11, line 1):

In Norway and Iceland, one self-developed question will be used instead of the 10-item Euthanasia scale. No questions will be asked about euthanasia in Germany. *Researchers from these countries were concerned that study participants would become anxious by these questions.*

8. There are many measurement instruments, what is the expected time to complete all these measurements for each participant? If the participant is not able to complete all the measurement at one time, are there any alternative methods could help? If a patient is in the terminal stage, he/she may become weaker day by day and the follow-up time of 4 weeks may not be applicable to some of these patients.

Response to the comment:

We have added details on completing the questionnaire. In the discussion we have addressed the last comment as a challenge of this study.

New text (Methods, p. 8, line 23):

Completing the questionnaire will take approximately 30-45 minutes. In the online version of the questionnaire, participants are allowed to save their answers and continue at a later time point. The same is applicable to completing the paper version of the questionnaire and the face-to-face interview.

New text (Discussion, p. 17, line 26):

Another challenge is that persons at the end of life may not be able to complete the follow-up questionnaire as they may become weaker over time. This will be monitored during the study and necessary actions will be taken in order to improve completion of the follow-up questionnaire.

Sample size

1. Please explain how the total sample size was estimated and how many participants will be recruited in each setting.

Response to the comment:

We have added details to the explanation of the sample size estimation.

New text (Methods, p. 12, line 13):

The primary outcomes are measured at baseline and 4 weeks post-inclusion. It is expected that 30% of all patients who complete the baseline assessment will be lost to follow-up, due to death, significant deterioration of health, or other causes. In that case, 70% of patients who complete the baseline measurement will be able to complete the assessment after 4 weeks at follow-up 1. Further, it is expected that 80% of all patients who complete the baseline assessment can be followed until death, whereas the remaining 20% are expected to either survive until the end of the data collection period or become lost to follow-up. Regarding the relatives, it is expected that in case patients who complete the baseline assessment die during follow-up, half of the bereaved relatives (i.e. 40% of all baseline patients), will be willing to complete a post-bereavement questionnaire (follow-up 2). The total cohort would thus include 2200 patients (n=200 per country) at baseline, 1540 patients (n=140 per country) at follow-up assessment 1, and 880 bereaved relatives (n=80 per country). *The number of 200 patients per country enables us to estimate proportions with 95% confidence intervals of approximately $\pm 7\%$. The number of recruiting sites will vary from two to six per country.*

Analyses

1. Please cite references for “Multivariate Imputation by Chained Equations (MICE).....” and please explain the reasons.

New text (Methods, p. 13, line 6):

Multivariate Imputation by Chained Equations (MICE) will be used to handle missing data (30), as we expect that patients may not be able or want to fill in all questions in the questionnaire. MICE is known to be a flexible principled method of addressing missing data and can handle variables of varying types (e.g. continuous or binary).

2. What statistical software will be used to analyze quantitative and qualitative data, respectively?

New text (Methods, p. 13, line 10)

Quantitative analyses will be performed with SPSS 25.0 statistical software.

New text (Methods, p. 14, line 7):

Data from the interviews will be imported into NVivo software for analysis.

3. Please add detailed descriptions for the descriptive and inferential statistics for the primary and secondary and other outcomes.

Response to the comment:

We have added details to the description of the descriptive and inferential statistics for the primary and secondary outcomes.

New text (Methods, p. 12, line 22):

Primary outcomes

The primary outcomes are experiences, concerns, expectations and preferences around dying and end-of-life care of patients in the last phase of life and their relatives, at baseline and after 4 weeks follow-up, and will be described in frequencies and narrative descriptions. *The proportion of patients who have certain concerns, expectations and preferences will be described. Sub group analyses will be performed to assess cross-gender, cross-age and cross-cultural variety on experiences, concerns, expectations and preferences. Narrative descriptions will be translated into English and categorized into themes that will be identified within the data.*

Descriptive statistics will be used to summarize *baseline* characteristics of the study participants (age, gender, education, *diagnosis, comorbidities*, religion, socioeconomic status, marital status, place of residence, quality of life, symptom load) by country and site. *Statistics on mean/median scores and variance will be presented where applicable.* Associations with country and patient characteristics will

be analysed in a multilevel modelling approach, taking account of clustering effects at country level. *Both univariable and multivariable analyses will be performed.* Repeated measures analyses of variance will be conducted to assess the development of outcomes between baseline and 4 weeks follow-up. All statistical tests will be two-sided and considered significant if $p < 0.05$. Multivariate Imputation by Chained Equations (MICE) will be used to handle missing data. *Quantitative analyses will be performed with SPSS 25.0 statistical software. Qualitative data from the interviews will be imported into NVivo software for analysis.*

Secondary outcomes

Secondary outcomes for patients include symptom load, HRQoL and wellbeing, and attitudes towards physician-assistance in dying. Secondary outcomes for relatives include HRQoL, well-being, informal care provision, attitudes towards physician-assistance in dying and bereavement. *The prevalence of these outcomes will be described in frequencies, mean/median scores and variance. Associations with country and patient characteristics will be analysed in a multilevel modelling approach, taking account of clustering effects at country level. Both univariable and multivariable analyses will be performed. Repeated measures analyses of variance will be conducted to assess the development of outcomes between baseline and 4 weeks follow-up.* The relationship of the relative to the patient will be taken into account in multivariable models, in addition to the characteristics mentioned for the analysis of the primary outcome.

VERSION 2 – REVIEW

REVIEWER	Dominic Wilkinson University of Oxford, Oxford Uehiro Centre for Practical Ethics
REVIEW RETURNED	27-Apr-2022
GENERAL COMMENTS	Thank you - my previous comments have been addressed.
REVIEWER	Hsiao-Ting Chang Taipei Veterans General Hospital, Department of Family Medicine
REVIEW RETURNED	11-May-2022
GENERAL COMMENTS	Thank you for the authors to provide detailed responses to my comments.